# Characterization of a Novel Rat Hepatitis E Virus Isolated from an Asian Musk Shrew (*Suncus murinus*)

**DOI:** 10.3390/v12070715

**Published:** 2020-07-01

**Authors:** Huimin Bai, Wei Li, Dawei Guan, Juan Su, Changwen Ke, Yasushi Ami, Yuriko Suzaki, Naokazu Takeda, Masamichi Muramatsu, Tian-Cheng Li

**Affiliations:** 1Department of Basic Medicine and Forensic Medicine, Baotou Medical College, Jianshe Road 31, Baotou 014060, China; baihuimin16@163.com; 2Institute of Microbiology, Center for Disease Control and Prevention of Guangdong Province, 160 Qunxian Road, Dashi Street, Panyu District, Guangzhou 511430, China; li_wei1994@163.com (W.L.); davidguan811@yahoo.com.cn (D.G.); susu05@126.com (J.S.); kechangwen@cdcp.org.cn (C.K.); 3Division of Experimental Animals Research, National Institute of Infectious Diseases, Gakuen 4-7-1, Musashi-murayama, Tokyo 208-0011, Japan; yami@nih.go.jp (Y.A.); ysuzaki@nih.go.jp (Y.S.); 4Research Institute for Microbial Diseases, Osaka University, Suita, Osaka 565-0781, Japan; seishunaotake@gmail.com; 5Department of Virology II, 2, National Institute of Infectious Diseases, Gakuen 4-7-1, Musashi-murayama, Tokyo 208-0011, Japan; muramatsu@nih.go.jp

**Keywords:** Asian musk shrew, rat HEV, nude rat, classification, subtype

## Abstract

The Asian musk shrew (shrew) is a new reservoir of a rat hepatitis E virus (HEV) that has been classified into genotype HEV-C1 in the species *Orthohepevirus C.* However, there is no information regarding classification of the new rat HEV based on the entire genome sequences, and it remains unclear whether rat HEV transmits from shrews to humans. We herein inoculated nude rats (Long-Evans rnu/rnu) with a serum sample from a shrew trapped in China, which was positive for rat HEV RNA, to isolate and characterize the rat HEV distributed in shrews. A rat HEV strain, S1129, was recovered from feces of the infected nude rat, indicating that rat HEV was capable of replicating in rats. S1129 adapted and grew well in PLC/PRF/5 cells, and the recovered virus (S1129c1) infected Wistar rats. The entire genomes of S1129 and S1129c1 contain four open reading frames and share 78.3–81.8% of the nucleotide sequence identities with known rat HEV isolates, demonstrating that rat HEVs are genetically diverse. We proposed that genotype HEV-C1 be further classified into subtypes HEV-C1a to HEV-C1d and that the S1129 strain circulating in the shrew belonged to the new subtype HEV-C1d. Further studies should focus on whether the S1129 strain infects humans.

## 1. Introduction

Hepatitis E virus (HEV) is a non-enveloped virus containing a positive-sense, single-strand RNA as the genome, and has been classified into the family *Hepeviridae*, which includes two genera, *Orthohepevirus* and *Piscihepevirus* [1]. The *Orthohepevirus* genus includes four species, *Orthohepevirus* A to D (HEV-A to HEV-D). HEV-A includes eight genotypes of HEV (G1 to G8 HEV), which are detected in humans, monkeys, pigs, wild boars, deer, mongooses, rabbits, and camels [1,2,3,4,5,6,7,8,9]. HEV-B includes avian HEVs exclusively, HEV-D includes several bat HEV strains [10,11]. HEV-C is grouped into two genotypes, HEV-C1, which includes rat HEV, and HEV-C2, which includes ferret HEV [12,13]. In addition to HEV-C1 and -C2, two putative genotypes, HEV-C3 and -C4, were recently found in rodents and kestrels [14,15].

Most reported cases of hepatitis E in humans were caused by HEV strains in HEV-A, such as G1 to G4 and G7 HEVs. However, recent reports demonstrated that rat HEV induced persistent HEV infection in a liver transplant recipient, and caused severe acute hepatitis E in an immunocompetent patient [16,17], and rat HEV RNA was detected in 6 of 2201 (0.27%) patients with hepatitis and 1 of 659 (0.15%) immunocompromised persons [18]. These results provided strong evidence that rat HEV is a potential causative agent for zoonotic infection.

Rat HEV was first identified in the feces of rats in Germany in 2010 [12]. Since then, rat HEV has been detected in several European and Asian countries and the United States of America (USA) [19,20,21,22,23]. The rat HEV genome contains three open reading frames (ORFs)—ORF1, ORF2, and ORF3—encoding a non-structural polyprotein, a capsid protein, and a small phosphoprotein, respectively, which is a common genome organization shared among all HEV-related viruses [12]. In addition, the rat HEV genome has a small ORF4 overlapping a part of ORF1, although its function is unknown [12,24]. Rat HEV shares only 50% to 60% nucleotide sequence identities with the HEV-A strains [12,24].

The primary host animals of rat HEV are thought to be rat species (*Rattus norvegicus*, *Rattus rattus*, and others), but the rat HEV RNA sequences have also been detected in the greater bandicoots (*Bandicota indica*) [20,25]. In addition, we have shown that rat HEV infection occurs in Asian musk shrews (*Suncus murinus*) (shrew), and that shrew is a new reservoir for rat HEV [26]. In that study we found that 10.4% (27/260) of the serum samples from shrews were positive for anti-rat HEV IgG antibody, 4.6% (12/260) were positive for IgM antibody, and 5 were positive for rat HEV RNA, and we determined the 281 nucleotide sequence corresponding to the C-terminal portion of ORF1 (GenBank accession nos. KC473527–KC473531) [26]. Although partial rat HEV genome sequences were recently detected in shrews in China [27,28], there is no information for the classification of the viruses using the entire genome sequences.

In the present study, we inoculated a nude rat with a serum sample from a shrew that was positive for rat HEV RNA. We found that the nude rat was susceptible to the rat HEV, and secreted it into the feces. The nucleotide sequence of the entire genome determined by next-generation sequencing analysis followed by phylogenetic analysis indicated that this strain was classified into a novel subtype of the rat HEV belonging to the genotype HEV-C1, and this virus was capable of replicating in a human hepatocarcinoma cell line, PLC/PRF/5, a fact which should permit its further genetic and antigenic characterization.

## 2. Materials and Methods

### 2.1. A Serum Sample from a Shrew

A blood sample was collected from a female shrew trapped in the Mazhang District of Zhanjiang City, Guangdong Province, China, in September 2012 [26]. The serum was separated by centrifugation at 2500× *g* for 20 min at 4 °C and stored at −80 °C until use. The serum sample was negative for hantavirus RNA and anti-rat HEV IgG antibody, and positive for anti-IgM antibody and rat HEV RNA (3.1 × 10^5^ copies/mL). The 281 nucleotide sequences corresponding to the C-terminal portion of ORF1 were determined (KC473530) [26].

### 2.2. Inoculation of Rats and Sample Collection

A nude rat (Long-Evans-rnu/rnu; Japan SLC Inc., Shizuoka, Japan) and two Wistar rats (Japan SLC Inc., Shizuoka, Japan) were used for the inoculation of rat HEV in this study. All of the rats were negative for rat HEV RNA and anti-rat HEV antibodies by a nested broad-spectrum RT-PCR and an enzyme-linked immunosorbent assay (ELISA), respectively. The nude rat was inoculated intravenously with the serum sample through the tail vein, and the Wistar rats were inoculated with the rat HEV-infected cell culture supernatant (see Section 2.3. Cell Culture and Virus Inoculation) in the same manner. Serum samples were collected weekly, and fecal samples were collected 2 times/week. The serum and fecal samples of the nude rat and the fecal samples of Wistar rats were used to examine the rat HEV RNA, and serum samples of Wistar rats were used for the detection of the anti-rat HEV IgG antibody.

The fecal samples were diluted with 10 mM phosphate-buffered saline (PBS) and shaken at 4 °C for 1 h. The 10% suspensions were clarified by centrifugation at 10,000 × g for 30 min and then passed through a 0.45 µm membrane filter (Millipore, Bedford, MA, USA). The rat experiments were reviewed and approved by the institutional ethics committee and were carried out according to the “Guides for Animal Experiments at the National Institute of Infectious Diseases, Tokyo, Japan” under code 112011 and 114012. Rats were individually housed in Biosafety Level-2 facilities.

### 2.3. Cell Culture and Virus Inoculation

A human hepatocarcinoma cell line, PLC/PRF/5 (JCRB0406), was obtained from the Health Science Research Resources Bank, and grown in Dulbecco’s modified Eagle’s medium (DMEM) supplemented with 10% (*v*/*v*) heat-inactivated fetal bovine serum (FBS; Nichirei Biosciences Inc., Tokyo, Japan), 100 U penicillin, and 100 mg streptomycin (Gibco, Grand Island, New York, NY, USA) at 37 °C in a humidified 5% CO_2_ atmosphere. The confluent cells were trypsinized and cultured in a 25 cm^2^ culture bottle (5 × 10^5^ cells/mL). The next day, 2 mL of 10% fecal suspension from the nude rat containing 5.2 × 10^8^ copies/mL of the RNA or the culture supernatant from the infected PLC/PRF/5 cells containing 5.0 × 10^7^ to 6.6 × 10^7^ copies/mL of the RNA was inoculated onto PLC/PRF/5 cells. After adsorption at 37 °C for 1 h, the cells were washed two times with PBS, then supplemented with 10 mL maintenance medium consisting of medium 199 (Invitrogen, Carlsbad, CA, USA) containing 2% (*v*/*v*) heat-inactivated FBS and 10 mM MgCl_2_. Further incubation was done at 36 °C. The culture medium was replaced with new medium every 4 days and used for detection of the capsid protein and RNA.

### 2.4. Detection of Rat HEV RNA

The viral RNA was extracted using the MagNA Pure LC system with a MagNA Pure LC Total Nucleic Acid Isolation Kit (Roche Applied Science, Mannheim, Germany) according to the manufacturer’s recommendations. A nested broad-spectrum RT-PCR analysis was performed to amplify a portion of ORF1, as described previously [29]. A one-step quantitative real-time-PCR (RT-qPCR) was carried out with a 7500 FAST Real-Time PCR System (Applied Biosystems, Foster City, CA, USA) using TaqMan Fast Virus 1-step Master Mix (Applied Biosystems). The RT-qPCR was performed under the following conditions: 5 min at 50 °C, 20 s incubation at 95 °C, followed by 40 cycles of 3 s at 95 °C and 30 s at 60 °C using the primer pair of 5′-CCACGGGGGTTAATACTGC-3′ (36–54) (sense) and 5′-CGGATGCGACCAAGAAACAG-3′ (189–208) (antisense) and the probe 5′-FAM-CGGCTACCGCCTTTGCTAATGC-TAMRA-3′ (81–102) [30]. The standard RNA of the RT-qPCR is the full genome of rat HEV (rat/R63/DEU/2009, R63) (GU345042) that was synthesized by using an mMESSAGE mMACHINE™ T7 Transcription Kit (Applied Biosystems). A 10-fold serial dilution of the standard RNA (10^7^ to 10^1^ copies) was used to quantify the viral copy numbers. Amplification data were collected and analyzed with Sequence Detector software version 1.3 (Applied Biosystems). The sensitivity of the RT-qPCR system was 10^3^ copies/mL.

### 2.5. Detection of IgG Antibodies and Rat HEV Antigen

Anti-rat HEV IgG antibodies were detected by ELISA using rat HEV-like particles, as described previously [31]. Antigen capture ELISA was used to detect the rat HEV capsid protein. Briefly, duplicate wells of flat-bottom 96-well polystyrene microplates (Dynex Technologies Inc., Chantilly, VA, USA) were coated with 100 µL of a coating buffer (0.1 M carbonate-bicarbonate buffer, pH 9.6) containing 1:2000 diluted serum from a rat HEV-LPs-immunized rabbit. The coating was performed at 4 °C overnight. Unbound antibodies were removed, the wells were washed twice with 10 mM PBS containing 0.05% Tween 20 (PBS-T), and then the blocking was carried out at 37 °C for 1 h with 150 µL of 5% skim milk (Difco Laboratories, Detroit, MI, USA) in PBS-T. A 100 µL aliquot of the cell culture supernatants was added to the wells and incubated for 1 h at 37 °C. After the wells were washed 3 times with PBS-T, 100 µL of guinea pig anti-rat HEV-LPs hyperimmune serum (1:2000 dilution with PBS-T containing 1% skim milk) was added to the wells, and the plate was incubated for 1 h at 37 °C. The plate was washed 3 times with PBS-T, and then horseradish peroxidase-conjugated goat anti-guinea pig IgG antibody (1:2000 in PBS-T containing 1% skim milk) (Cappel, Durham, NC, USA) was added to each well. After incubation for 1 h at 37 °C, the plate was washed 3 times with PBS-T and 100 µL of substrate solution containing o-phenylenediamine and H_2_O_2_ was added. The plate was left for 30 min at room temperature, and then the reaction was stopped with 50 µL of 4 N H_2_SO_4_. After 10 min, the absorbance at 492 nm was measured with a microplate reader (Molecular Devices Corp., Tokyo, Japan). The normal cell culture supernatant (three wells per plate) served as the negative control. When the ratio of the optical density (OD) values between the sample and negative control was higher than 3.0, the sample was judged to be positive.

### 2.6. Next-Generation Sequencing Analyses (NGS)

The RNA extraction was described above. A 200 bp fragment library was constructed for each sample using the NEBNext Ultra RNA Library Prep Kit for Illumina v1.2 (New England Biolabs, Ipswich, MA, USA) according to the manufacturer’s instructions. Samples were barcoded for multiplexing using NEBNext Multiplex Oligos for Illumina Index Primer Sets 1 and 2 (New England Biolabs, Ipswich, MA, USA). Library purification was done using Agencourt AMPure XP magnetic beads (Beckman Coulter, Pasadena, CA, USA), as recommended in the NEBNext protocol. The quality of the purified libraries was assessed on an MCE-202 MultiNA (Shimadzu Corporation, Kyoto, Japan), and the concentrations were determined on a Qubit 2.0 fluorometer using the Qubit HS DNA Assay (Invitrogen, Carlsbad, CA, USA). A 151-cycle paired-end read sequencing run was carried out on a MiSeq desktop sequencer (Illumina, San Diego, CA, USA) using the MiSeq Reagent Kit v2 (300 cycles). Following preliminary analysis, the MiSeq reporter program was used to generate FASTQ formatted sequence data for each sample. Sequence data were analyzed using CLC Genomics Workbench Software v6.5.1 (CLC Bio, Aarhus, Denmark). Contigs were assembled from the obtained sequence reads by de novo assembly. Subsequently, the assembled contig sequences were used to query the non-redundant nucleotide database in GenBank by employing the BLAST. In addition, the 3′-terminal sequence was further amplified by PCR with the primers 1129F6521 (5′-TCAGGCCTGwGTGTACTACCA-3′) and TX30SXN (5′-GACTAGTTCTAGATCGCGAGCGGCCGCCCTTTTTTTTTTTTTTTTTTTTTTTTTT-3′), and the sequence was analyzed by using the primer 1129F6521.

### 2.7. Phylogenetic Analyses

Phylogenetic trees with 1000 bootstrap replicates were generated by the neighbor-joining method based on the entire genome of HEVs. The nucleotide sequence alignment was performed using Clustal X 1.81. The genetic distance was calculated by Kimura’s two-parameter method [32].

### 2.8. Examination of Alanine Aminotransferase (ALT)

ALT values in Wistar rats were monitored weekly using a Fuji Dri-Chem Slide GPT/ALT-PIII kit (Fujifilm, Saitama, Japan). The geometric mean titers of ALT over the preinoculation period were defined as normal ALT, and a 2-fold or greater increase at the peak was considered a sign of hepatitis [33].

## 3. Results

### 3.1. Isolation of a Rat HEV Strain from a Shrew

To isolate rat HEV from the shrew, 100 μL of the serum from the shrew was diluted with 300 μL of PBS and intravenously inoculated into a 17-week-old female nude rat through the tail vein. The virus replication was monitored, as shown in Figure 1. The viral RNA was detected in the feces on day 8 post-inoculation (p.i.), counting a copy number of 2.82 × 10^4^ copies/g, and in the sera on day 14 p.i., counting a copy number of 1.38 × 10^3^ copies/mL. Then the RNA increased to 5.23 × 10^9^ copy/g in the feces and 1.68 × 10^5^ copy/mL in the sera on day 35 p.i. A partial sequence (281 nucleotides) of ORF1 was amplified using the serum and fecal samples collected on day 35 p.i. by RT-PCR, and the nucleotide sequence was determined. It was identical to that of CHZ-sRatE-1129 (KC473530), which was detected in the same shrew in our previous study [26]. These results indicated that the shrew serum contained replication-competent rat HEV. The rat HEV strain was named S1129.

### 3.2. Growth of the S1129 Strain in PLC/PRF/5 Cells

To examine whether the S1129 strain grows in the PLC/PRF/5 cells, 2 mL of the 10% suspension of the fecal specimen collected from the nude rat on day 35 p.i. was inoculated onto PLC/PRF/5 cells (5 × 10^5^ cells). The viral RNA was detected in the culture supernatant on day 64 p.i., with a copy number of 1.82 × 10^3^ copies/mL, then gradually increased, reaching 4.61 × 10^7^ copies/mL on day 172 p.i. (Figure 2A). The capsid protein was detected on day 120 p.i. with an OD value of 0.119, increased gradually, and reached OD 0.515 on day 172 p.i. (Figure 2B). Extensive virus growth was constantly observed in the second and third passage by an earlier detection of both viral RNA and the capsid antigen in the culture medium (Figure 2), indicating that the S1129 strain was able to replicate in the PLC/PRF/5 cells. The rat HEV strains recovered from the cell culture were named S1129c1, S1129c2, and S1129c3, according to the passage number.

### 3.3. Infectivity of Rat HEV In Vivo

To examine whether the rat HEV recovered from the cell culture was capable of infecting ordinary rats, 500 μL of the culture supernatant containing 5 × 10^7^ copies/mL of S1129c1 collected on day 168 p.i. was intravenously inoculated into two 15-week-old female Wistar rats (WR1 and WR2). The virus replication was monitored, as depicted in Figure 3. The viral RNA in the feces was first detected on day 7 p.i., with copy numbers of 1.14 × 10^4^ copies/g and 3.56 × 10^4^ copies/g in the two animals, and reached a peak on day 21 p.i., with copy numbers of 1.45 × 10^7^ copies/g and 6.14 × 10^4^ copies/g, respectively. Then the RNA was decreased and became undetectable after day 35 p.i. in both rats. Anti-rat HEV IgG antibodies were detected on day 7 p.i. in both rats, and reached a peak on day 21 p.i., with OD values of 2.785 and 2.998, respectively. These results indicated that the rat HEV recovered from the cell culture was infectious to normal rats. During the period of the infection experiment, no elevation of ALT was observed in the two rats, demonstrating that rat HEV infection did not induce liver damage in rats.

### 3.4. Characterization of Full Genome Sequences of S1129 and S1129c1

The entire genome sequences of the S1129 strain isolated from the infected nude rat (accession no. LC549186) and the S1129c1strain recovered from the infected cell culture supernatant (LC549187) were analyzed by next-generation sequencing (NGS). Both the S1129 and S1129c1 strains consisted of 6960 nucleotides plus an undetermined length of poly (A) tail. The 5′ terminal untranslated region (5′ UTR) and 3′ UTR contained 10 (nucleotide sequence numbers 1–10) and 65 nucleotides (6896–6960), respectively. Both strains contained four ORFs, ORF1 containing 4923 nucleotides (11–4933), ORF4 containing 552 (27–578), ORF3 containing 309 nucleotides (4950–5258), and ORF2 containing 1935 nucleotides (4961–6895). At present, a total of 22 full or near-full genome sequences of rat HEV are available in the GenBank/DDBJ. The number of amino acid residues of ORF1 was variable (1629–1642 amino acids) due to deletions or insertions in the hypervariable region (amino acid residues 831 to 883 corresponding to the Indonesian strain AB847305). ORF1 of both strains encodes 1640 amino acids, which is the same as that of Indonesian strain AB847309 and Vietnamese strain MG600417, but shorter than Indonesian strains AB847305, AB847306, LC145325, LC225388, and LC225389, and longer than the remaining 15 rat HEV strains. ORF1 of both strains possessed two amino acid deletions at residues 600 (Tyr) and 841 (Pro) when compared with the above Indonesian strain AB847305. ORF2 and ORF3 encode 644 and 102 amino acids, respectively, as do other rat HEV strains. However, the ORF4 of S1129 and S1129C1 encodes 183 amino acids, which is the same number as encoded by the ORF4 of German strains R63 (GU345042) and R68 (GU345043), and the putative ORF5 and ORF6 were not found in S1129 and S1129c1, suggesting these two ORFs are not common in rat HEV.

The S1129 strain grew in the PLC/PRF/5 cells and produced an infectious rat HEV, S1129c1. However, we found a total of 32 nucleotide differences between the two strains, leading to four amino acid mutations in ORF1, seven amino acid mutations in ORF2, two amino acid mutations in ORF3, and one amino acid mutation in ORF4 (Table 1).

The nucleotide sequence identities between S1129 and G1 to G8, rabbit, rat, ferret, kestrel, bat, and avian HEV strains were compared. S1129 shared 46.1% to 81.8% of the nucleotide sequence identities with known HEV strains and was most close to HEV-C1 with identities of 78.3% to 81.8%. In each ORF, the nucleotide and amino acid sequence identities between S1129 and HEV-C1 were higher than those between S1129 and other HEVs (HEV-A, -B, -C2, -C3, -C4 and -D) (Table 2).

### 3.5. S1129 Belongs to a New Subtype of HEV-C1

A phylogenetic tree was generated based on the entire or nearly entire genome of the prototype of G1 to G8, bat, avian, ferret, kestrel, and rat HEV strains available in GenBank/DDBJ (Figure 4). The genetic tree showed that all 24 of the rat HEV strains could be separated into four genetic groups based on the high bootstrap values of nearly 100%. The rat HEV has been classified as genotype HEV-C1 [1], and we would like to propose that genotype HEV-C1 was further genetically segregated into four subtypes, HEV-C1a to HEV-C1d. HEV-C1a contained 9 strains from Germany, the USA, Indonesia, China, and Canada; HEV-C1b included 11 strains from Indonesia, China, and Vietnam; and HEV-C1c included 2 strains from Indonesia. The S1129 and S1129c1 strains formed a novel subtype of HEV-C1, HEV-C1d, which was distinctly separated from all known rat HEV strains. These results suggested that a new subtype of rat HEV is circulating in shrews in China (Figure 4).

## 4. Discussion

In the present study, we isolated a novel rat HEV strain, S1129, through in vivo amplification by inoculating a nude rat with a serum sample derived from a rat HEV RNA-positive shrew trapped in China. Phylogenetic analysis demonstrated that this strain was highly divergently constellated in known rat HEV isolates, and we proposed that HEV-C1 was further classified into subtypes HEV-C1a to HEV-C1d, and the S1129 and S1129c1 strains belonged to a new subtype, HEV-C1d (Figure 4). Interestingly, we found that the subtype HEV-C1a included the isolates from the USA, Germany, Indonesia, China, and Canada, and HEV-C1b included the isolates from Vietnam, China, and Indonesia, suggesting that genetically diverged strains of rat HEV are globally distributed. However, due to the limited complete sequence information, we did not find a correlation between the subtypes and the geographical distribution of rat HEV.

Although S1129 grew in the PLC/PRF/5 cells, the viral RNA became detectable on day 64 p.i., more than 2 months after the inoculation. In contrast, S1129c1 began to grow exponentially on day 12 after the incubation, indicating that S1129c1 was more adapted to the PLC/PRF/5 cells. We found 32 nucleotide changes after a long-term incubation in the non-host animal-derived cells. Therefore, some of the amino acid mutations led the rat HEV to adapt to the PLC/PRF/5 cells. After adaptation in PLC/PRF/5 cells, the nucleotide exchanges were also found in the strain R63 [30], but no common adaptation mutations were found between R63 and 1129c1. In the future, it would be of interest to analyze the relationship between the replication period and the mutations detected in the viral genome. A reverse genetics system for rat HEV will be useful to clarify the key nucleotide changes responsible for the adaptation.

Considering the long incubation period and the resultant accumulation of mutations, cell culture may not be the best way to isolate rat HEV, especially when the samples contain low titers of virus. In contrast, nude rats have been shown to be highly susceptible to rat HEV infection, while shedding high titers of rat HEV for a prolonged period, and the viral genome is stable during the replication in nude rats [30,34]. Therefore, nude rats would be suitable for the isolation of rat HEV.

Before the recent reports of hepatitis cases in humans, there had been no clear evidence of the zoonotic potential of rat HEV infection. Indeed, the experimental infections of rhesus monkeys and pigs with rat HEV did not lead to any signs of virus replication [21,35]. Serological analyses of human and porcine sera demonstrated that fewer sera contained anti-rat HEV-specific antibodies compared with human HEVs, suggesting that transmission of rat HEV-related viruses to humans or pigs is a rare event [36,37]. However, several recent cases of hepatitis E appeared to have been caused by rat HEV infection, clearly showing the possibility of zoonotic infection due to rat HEV [16,17,18]. To date, two entire genomes of rat HEV detected in the patients have been analyzed. One of the strains, MG813927/Hong Kong/Human, belonged to HEV-C1b and was detected in a patient in Hong Kong; this strain possessed as high as 93.9% nucleotide sequence identity with a rat HEV (JX120573/Vietnam) detected in Vietnam [16,24]. The other strain, MK050105/Canada/Human, belonged to HEV-C1a and was detected in a patient in Canada [17] (Figure 4).

Although a USA strain, KM516906/USA, belonging to HEV-C1a did not infect rhesus monkeys [21], a Canadian strain, MK050105/Canada/Human, included in the same isolated HEV-C1a was isolated from an acute hepatitis patient [17]. It remains unclear whether all rat HEV infects humans, or whether limited kinds of rat HEV subtypes or strains are capable of infecting humans. Experiments using non-human primates and the HEV-C1d strain (S1129) obtained in this study may provide a clue to clarify this point.

As shown in Table 2, the amino acid identities between S1129 and other rat HEV isolates were 86.8–90.6% in ORF1 and 66.7–79.4% in ORF3. In contrast, relatively high amino acid identities (92.2–95.5%) were found in ORF2 between S1129 and rat HEV isolates, suggesting that rat HEV shares similar antigenicity. In fact, rat HEV-like particles derived from R63 (GU42052) were reactive to the serum from S1129-infected Wistar rat, and these particles were used to detect the anti-rat HEV IgG and IgM antibodies in Vietnamese rats [31].

In consideration of the mounting evidence of zoonotic infection due to rat HEV, there is an urgent need for specific molecular and serological assays for the diagnosis of hepatitis E caused by HEV-C1. Currently, the commercially available ELISA kits for the detection of HEV antibodies use an antigen derived from HEV-A. Because the amino acid identities in ORF2 between HEV-C1 and HEV-A are less than 60%, and anti-rat HEV antibody has low cross-reactivity with G1, G3, and G4 HEV-like particles [12,31], an ELISA based on an antigen from HEV-A would seem to lack sensitivity for the detection of anti-rat HEV antibodies. The establishment of a method to detect rat HEV IgG and IgM antibodies using rat HEV-like particles would thus be useful for the seroepidemiology and diagnosis of hepatitis E caused by rat HEV infection.

## Figures and Tables

**Figure 1 viruses-12-00715-f001:**
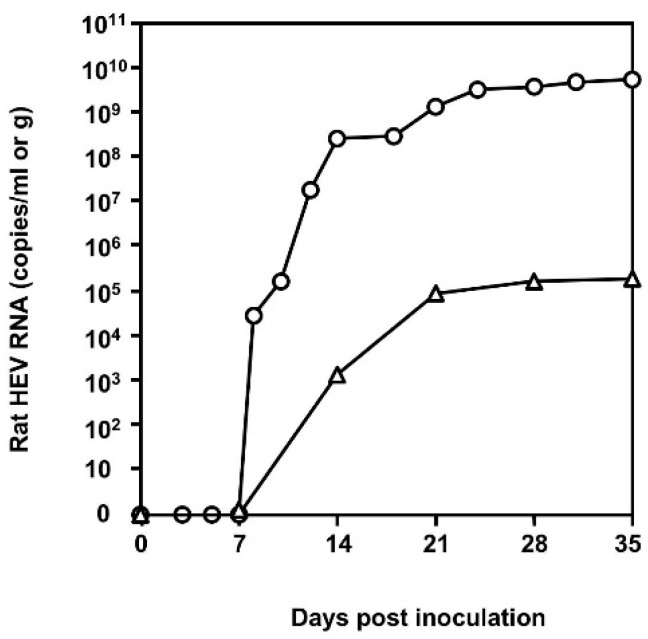
Kinetics of rat hepatitis E virus (HEV) RNA in the nude rat. A nude rat was inoculated with the serum collected from a shrew. HEV RNA in the feces (○) and sera (∆) was monitored by RT-qPCR.

**Figure 2 viruses-12-00715-f002:**
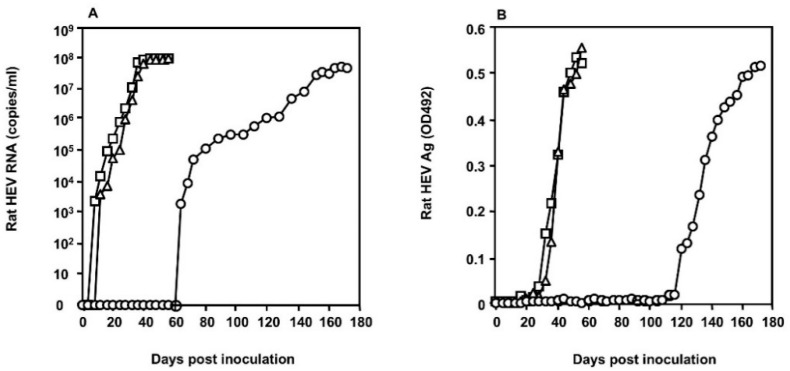
Replication of S1129 in the human hepatocarcinoma cell line PLC/PRF/5. Two mL of 10% fecal suspension was inoculated onto PLC/PRF/5 cells. The culture supernatants were collected every 4 days and used for the detection of rat HEV RNA by RT-qPCR (**A**) and for the capsid antigen detected by rat HEV-specific antigen enzyme-linked immunosorbent assay (ELISA) (**B**). The first (S1129c1) (○), second (S1129c2) (∆), and third (S1129c3) (□) passages in the PLC/PRF/5 cells are shown.

**Figure 3 viruses-12-00715-f003:**
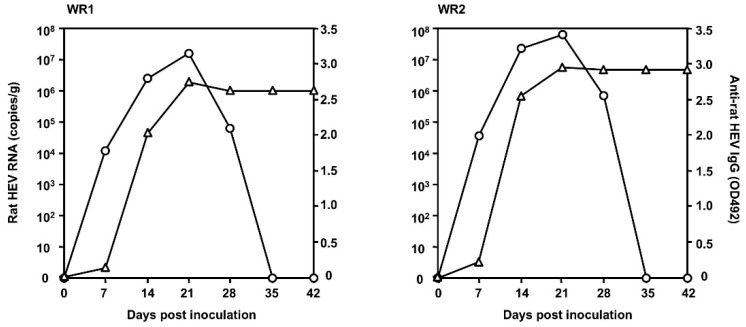
Infectivity of the rat HEV recovered from the culture supernatant. Two Wistar rats (WR1 and WR2) were inoculated with the S1129-infected culture supernatant collected on day 168. After inoculation, fecal samples were collected two times/week, and serum samples were collected weekly until 7 weeks. Rat HEV RNA in the stool was monitored by RT-qPCR (○), and anti-rat HEV IgG (∆) antibodies were detected by ELISA.

**Figure 4 viruses-12-00715-f004:**
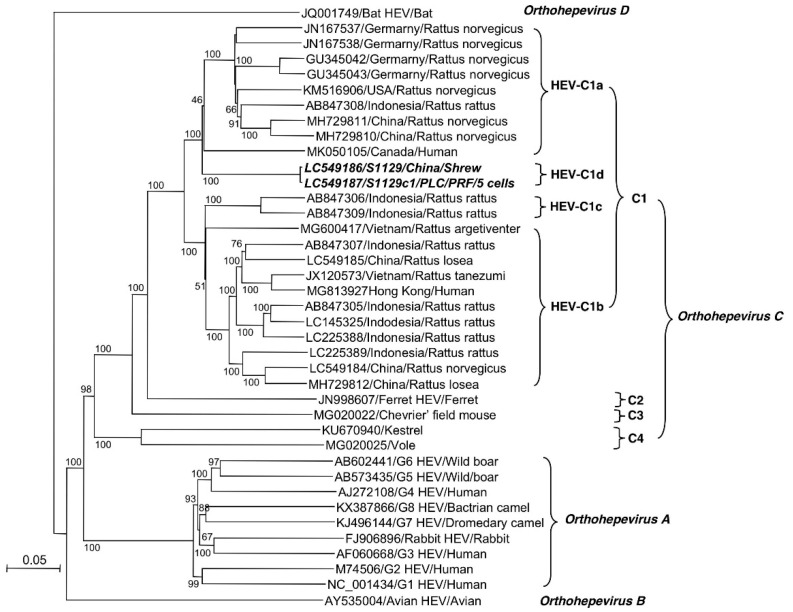
Phylogenetic analyses based on the entire genome. Phylogenetic trees with 1000 bootstrap replicates were generated by the neighbor-joining method based on the entire genome of HEVs. S1129 and S1129c1 detected in the present study are shown in bold italic letters.

**Table 1 viruses-12-00715-t001:** Nucleotide and amino acid changes in S1129c1 during passages.

Nucleotide			Amino Acid				
^a^ Position	S1129	S1129c1	^b^ Position	ORF1	ORF2	ORF3	ORF4
346	T	C	107				Val/Ala
766	T	C					
957	T	A	316	Phe/Tyr			
1639	C	T					
1671	C	T	534	Thr/Ile			
1732	T	C					
1756	C	T					
1822	T	C					
1907	C	T	633	His/Tyr			
2560	T	C					
2561	T	C	851	Ser/Pro			
2563	T	C					
2740	C	T					
2914	T	G					
2958	C	T					
3211	T	C					
3649	T	C					
3859	G	A					
4144	T	C					
4588	A	T					
5115	G	A	^c^ 52 (56)		Arg/Gln	Gly/Arg	
5156	C	T	66		Leu/Phe		
5179	C	T	77			Ala/Val	
5335	C	T					
5552	C	A	198		Leu/Met		
5634	T	C	225		Val/Ala		
5661	T	G	234		Val/Gly		
5869	C	T					
6624	C	T	555		Ser/Phe		
6695	C	T					
6762	C	T	579		Ser/Leu		
6908	T	C					

^a^ Nucleotide positions in the genomic RNA. ^b^ Amino acid positions in each ORF. ^c^ The number in parentheses indicates the aa position of ORF3.

**Table 2 viruses-12-00715-t002:** Nucleotide and deduced amino acid-sequence identities between S1129 and other hepatitis E virus (HEV) isolates.

Rat HEV S1129 (LC549186)								
		Nucleotides (%)				Amino Acids (%)		
HEV (Accession No)	Species	Entire Genome	ORF1	ORF2	ORF3	ORF1	ORF2	ORF3
Bat HEV (JQ001749)	HEV-D	**46.1**	**48.5**	55.2	**46.0**	50.5	52.8	50.0
Rat HEV (GU345042)	HEV-C1	80.9	79.4	84.3	**89.0 ^a^**	90.1	95.4	78.4
Rat HEV (GU345043)	HEV-C1	80.0	79.1	83.4	88.7	89.5	**95.5**	78.4
Rat HEV (JN167537)	HEV-C1	80.8	79.8	83.3	87.7	90.9	94.6	75.5
Rat HEV (AB847308)	HEV-C1	80.5	80.8	83.0	86.4	90.8	94.1	74.5
Rat HEV (KM516906)	HEV-C1	**81.8**	**81.0**	**83.5**	88.0	**90.6**	94.7	**79.4**
Rat HEV (AB847306)	HEV-C1	79.4	77.7	80.3	80.9	87.7	92.9	67.6
Rat HEV (AB847309)	HEV-C1	78.8	77.7	79.9	79.5	86.5	92.4	66.7
Rat HEV (AB847305)	HEV-C1	78.5	78.4	79.3	81.6	87.5	93.2	67.6
Rat HEV (AB847307)	HEV-C1	78.3	78.5	79.4	82.8	87.1	93.8	72.5
Rat HEV (JX120573)	HEV-C1	78.9	78.1	80.0	83.7	86.8	93.5	72.5
Rat HEV (MK050105)	HEV-C1	80.7	79.7	82.9	86.6	90.1	94.6	75.6
Ferret HEV (AB890001)	HEV-C2	60.8	56.4	71.1	63.7	73.9	79.1	45.6
Chevrier HEV(MG020022)	HEV-C3	64.9	63.4	68.3	65.1	66.8	74.3	46.1
Kestrel HEV (KU670940)	HEV-C4	54.0	57.2	62.4	46.9	54.2	69.1	29.2
Genotype 1 (NC-001434)	HEV-A	49.3	51.3	60.4	51.9	57.1	55.5	33.3
Genotype 2 (M74506)	HEV-A	49.0	50.5	58.8	52.6	57.0	55.2	30.0
Genotype 3 (AF060668)	HEV-A	47.3	51.0	60.1	49.4	57.3	56.8	26.6
Genotype 4 (AJ272108)	HEV-A	49.2	51.4	60.5	56.5	57.0	56.5	25.7
Genotype 5 (AB573435)	HEV-A	50.0	51.9	60.9	55.3	57.5	56.0	27.1
Genotype 6 (AB602441)	HEV-A	48.4	51.2	60.5	53.6	57.5	56.1	28.9
Genotype 7 (KJ496143)	HEV-A	48.2	50.2	58.5	50.3	57.5	56.6	**23.7**
Genotype 8 (KX387866)	HEV-A	48.3	48.2	60.5	51.6	57.6	55.6	25.0
Rabbit HEV (FJ906895)	HEV-A	48.9	51.1	58.8	51.4	51.8	56.1	27.6
Avian HEV (AY535004)	HEV-B	48.5	49.7	**54.2**	47.0	**44.7**	**47.4**	33.9

a: The highest and lowest identities are shown in bold letters.

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
