# Peer review of "Characterization of a Novel Rat Hepatitis E Virus Isolated from an Asian Musk Shrew (Suncus murinus)"

_viruses, 2020, doi:10.3390/v12070715_

Round 1
Reviewer 1 Report
In the presented manuscript, Bai and colleagues further characterize a rat hepatitis E virus isolated from Asian musk shrew that the group has publishes before. They propose it to be classified as a new subtype of HEV-C1.
The Asian musk shrew has been described before as a reservoir by the very same group, while rat HEV strains replicating in cell culture have also been described by some of the authors in another paper (“Construction and Characterization of an Infectious cDNA Clone of Rat Hepatitis E Virus”; DOI: 10.1099/vir.0.000072 ), even with a reverse genetic system. The fact that i.v. injection of rats with the recovered hepatitis E virus do lead to rat infection is technically novel, however, not very surprising. The finding that HEV-C1 might need to be further subclassified is a novel finding but does warrant further confirmation. To improve the present study, the following issues should be considered:
Additional remarks:
- Whenever the authors show qPCR data, they have to indicate their limit of detection in the figure
- Line 40-41 “both HEV-B and -C are non-zoonotic”. The authors mention later on that there HEV-C has been detected in humans so they should adapt this sentence accordingly
- The authors are able to infect PLC/PRF/5 cells with fecal derived rat HEV. It would be interesting to test additional hepatic cell lines such as HepG2 or Huh7.5.
- From their phylogenetic tree, the authors conclude that their newly introduced HEV C1 subtypes do not cluster geographically (line 291). These findings are surprising and authors should have discussed reasons, why there is no clear geographic clustering.
- Some of the authors published about a reverse genetic system previously (“Construction and Characterization of an Infectious cDNA Clone of Rat Hepatitis E Virus”; DOI: 1099/vir.0.000072 ). In this paper they also reported on nucleotide exchanges after cell culture adaptation. Are there similar adaptations as here? The authors should have discussed this in depth. If there are common adaptation mutations, they should test them in a reverse genetic system for replication advantages.
Author Response
Responses to Reviewer 1:
In the presented manuscript, Bai and colleagues further characterize a rat hepatitis E virus isolated from Asian musk shrew that the group has publishes before. They propose it to be classified as a new subtype of HEV-C1.
The Asian musk shrew has been described before as a reservoir by the very same group, while rat HEV strains replicating in cell culture have also been described by some of the authors in another paper (“Construction and Characterization of an Infectious cDNA Clone of Rat Hepatitis E Virus”; DOI: 10.1099/vir.0.000072 ), even with a reverse genetic system. The fact that i.v. injection of rats with the recovered hepatitis E virus do lead to rat infection is technically novel, however, not very surprising. The finding that HEV-C1 might need to be further subclassified is a novel finding but does warrant further confirmation. To improve the present study, the following issues should be considered:
Additional remarks:
- Whenever the authors show qPCR data, they have to indicate their limit of detection in the figure
Response:
Instead of indicating the limit of detection in the figure, we added the following description of the sensitivity of the RT-qPCR: “The sensitivity of the RT-qPCR system was 103 copies/ml.” (section 2.4.: Detection of rat HEV RNA).
- Line 40-41 “both HEV-B and -C are non-zoonotic”. The authors mention later on that there HEV-C has been detected in humans so they should adapt this sentence accordingly
Response:
We deleted the phrase “and both HEV-B and -C are non-zoonotic.”
- The authors are able to infect PLC/PRF/5 cells with fecal derived rat HEV. It would be interesting to test additional hepatic cell lines such as HepG2 or Huh7.5.
Response:
We agree with the reviewer. We would like to examine the replication of rat HEV isolated from Shrew in other hepatic cell lines in the future.
- From their phylogenetic tree, the authors conclude that their newly introduced HEV C1 subtypes do not cluster geographically (line 291). These findings are surprising and authors should have discussed reasons, why there is no clear geographic clustering.
Response:
We revised this sentence as follows: “However, due to the limited complete sequence information we did not find a correlation between the subtypes and the geographical distribution of rat HEV.” (page 19, lines 345-346)
- Some of the authors published about a reverse genetic system previously (“Construction and Characterization of an Infectious cDNA Clone of Rat Hepatitis E Virus”; DOI: 1099/vir.0.000072 ). In this paper they also reported on nucleotide exchanges after cell culture adaptation. Are there similar adaptations as here? The authors should have discussed this in depth. If there are common adaptation mutations, they should test them in a reverse genetic system for replication advantages.
Response:
According to the reviewer’s suggestion, we compared the mutations occurring in rat HEV strains R63 and 1129c1, but we did not find mutations at the same nucleotide positions. We added the following sentence: “After adaptation in PLC/PRF/5 cells, the nucleotide exchanges were also found in the strain R63 [30], but no common adaptation mutations were found between R63 and 1129c1.” (page 19, lines 352-354)
Reviewer 2 Report
In this paper, Bai et al. analyzed the characteristics of a rat HEV obtained form an Asian musk shrew. Previously the authors reported that the rat HEV infects Asian musk shrews in China, and in this study, they performed infection experiments using nude rats and a human hepatoma cell line. The results indicate that this rat HEV in Asian musk shrew has a potential to infect humans. Additionally, the entire genomic sequences of this rat HEV strain were determined and a new subtype of HEV-C1, HEV-C1d, is suggested. This study shows new important data for the characterization of Orthohepevirus genus. I have some minor comments to be addressed as below.
- Line 87, please show the full spelling for ‘PBS’ in the first appearance, instead of line 93.
- Line 166, please show the sequence of primer TX30SXN.
- Line 300, ‘ncubation’ should be corrected to ‘incubation’.
- Table 1, probably this is an editorial error but the positions of each index is inappropriate. Also, ‘Amino aci’ should be corrected to ‘Amino acid’.
Author Response
Responses to Reviewer 2:
Comments and Suggestions for Authors
In this paper, Bai et al. analyzed the characteristics of a rat HEV obtained form an Asian musk shrew. Previously the authors reported that the rat HEV infects Asian musk shrews in China, and in this study, they performed infection experiments using nude rats and a human hepatoma cell line. The results indicate that this rat HEV in Asian musk shrew has a potential to infect humans. Additionally, the entire genomic sequences of this rat HEV strain were determined and a new subtype of HEV-C1, HEV-C1d, is suggested. This study shows new important data for the characterization of Orthohepevirus genus. I have some minor comments to be addressed as below.
- Line 87, please show the full spelling for ‘PBS’ in the first appearance, instead of line 93.
Response:
Corrected.
- Line 166, please show the sequence of primer TX30SXN.
Response: The sequence of the primer TX30SXN was added on page 9, lines 200-201 in the revised manuscript.
- Line 300, ‘ncubation’ should be corrected to ‘incubation’.
Response:
Corrected.
- Table 1, probably this is an editorial error but the positions of each index is inappropriate. Also, ‘Amino aci’ should be corrected to ‘Amino acid’.
Response:
We confirmed that the positions of each index are appropriate in the uploaded file. “Amino aci” was corrected to “Amino acid”. 
Round 2
Reviewer 1 Report
Comments have been answered.